# Enhanced Anti-Bacterial Activity of Arachidonic Acid against the Cariogenic Bacterium *Streptococcus mutans* in Combination with Triclosan and Fluoride

**DOI:** 10.3390/antibiotics13060540

**Published:** 2024-06-10

**Authors:** Avraham Melkam, Ronit Vogt Sionov, Miriam Shalish, Doron Steinberg

**Affiliations:** 1Faculty of Dental Medicine, Ein Kerem Campus, Institute of Biomedical and Oral Research (IBOR), The Hebrew University of Jerusalem, Jerusalem 9112102, Israel; avraham.melkam@mail.huji.ac.il (A.M.); ronit.sionov@mail.huji.ac.il (R.V.S.); 2Hadassah Medical Center, Department of Orthodontics, Faculty of Dental Medicine, The Hebrew University of Jerusalem, Jerusalem 9112102, Israel; mshalish@mail.huji.ac.il

**Keywords:** anti-bacterial, anti-biofilm, arachidonic acid, chlorhexidine, cetylpyridinium chloride, fluoride, triclosan, *Streptococcus mutans*

## Abstract

Dental caries is a global health problem that requires better prevention measures. One of the goals is to reduce the prevalence of the cariogenic Gram-positive bacterium *Streptococcus mutans*. We have recently shown that naturally occurring arachidonic acid (AA) has both anti-bacterial and anti-biofilm activities against this bacterium. An important question is how these activities are affected by other anti-bacterial compounds commonly used in mouthwashes. Here, we studied the combined treatment of AA with chlorhexidine (CHX), cetylpyridinium chloride (CPC), triclosan, and fluoride. Checkerboard microtiter assays were performed to determine the effects on bacterial growth and viability. Biofilms were quantified using the MTT metabolic assay, crystal violet (CV) staining, and live/dead staining with SYTO 9/propidium iodide (PI) visualized by spinning disk confocal microscopy (SDCM). The bacterial morphology and the topography of the biofilms were visualized by high-resolution scanning electron microscopy (HR-SEM). The effect of selected drug combinations on cell viability and membrane potential was investigated by flow cytometry using SYTO 9/PI staining and the potentiometric dye DiOC2(3), respectively. We found that CHX and CPC had an antagonistic effect on AA at certain concentrations, while an additive effect was observed with triclosan and fluoride. This prompted us to investigate the triple treatment of AA, triclosan, and fluoride, which was more effective than either compound alone or the double treatment. We observed an increase in the percentage of PI-positive bacteria, indicating increased bacterial cell death. Only AA caused significant membrane hyperpolarization, which was not significantly enhanced by either triclosan or fluoride. In conclusion, our data suggest that AA can be used together with triclosan and fluoride to improve the efficacy of oral health care.

## 1. Introduction

*Streptococcus mutans* is a facultative anaerobic Gram-positive bacterium that is involved in the development of dental caries by virtue of its ability to produce acids such as lactic acid (acidogenicity), form biofilms on tooth enamel, and survive under acidic conditions (aciduricity) [1,2,3,4]. The acid, which is produced by the fermentation of sugars, results in demineralization of hydroxyapatite [2,3]. The critical pH value is 5.5, and prolonged periods below this value promote cavitations [2]. Sugars also induce the expression of genes involved in biofilm formation, including glucosyltransferases, glucan-binding proteins, collagen-binding proteins, and adhesins [1,4]. Within a biofilm, the bacteria are enwrapped in an extracellular matrix consisting of exopolysaccharides (EPSs) produced by the bacteria, salivary components such as minerals and proteins, and food debris. Oral biofilms are usually composed of several species of bacteria and fungi [4,5,6,7]. The bacteria in biofilms are usually sessile, have a low metabolic rate, and are protected from environmental stress stimuli, including antibiotics and alterations in pH [1,8,9]. Acidic stress triggers an acid tolerance response that, among others, stimulates the expression of several genes, including those encoding the F_1_F_0_-ATPase complex that pumps out protons, thereby preventing cytoplasmic acidification [2]. The protons expelled from the bacteria and the secreted acids become sequestered in the extracellular matrix of biofilms, which leads to local acidification and enamel decay [4]. The lowering of the pH value in the biofilm matrix also leads to dysbiosis with a preference for the growth of acid-tolerant microorganisms [4,10].

Dental caries is a global health problem affecting people of all ages with a high economic burden on society [10,11,12]. Untreated oral biofilm-associated infections may lead to periodontal diseases, tooth loss, and even systemic diseases such as pneumonia and endocarditis [13,14]. Many approaches are used to reduce the burden of *S. mutans* and other cariogenic bacteria in the oral cavity to improve oral health [10,15]. These include mechanical removal by tooth brushing and flossing, removal of dental plaque by dentists, fluoride-containing toothpastes, and various mouthwashes and other dentifrices containing antiseptic compounds such as chlorhexidine (CHX), cetylpyridinium chloride (CPC), and triclosan [16,17,18,19,20]. Other ingredients include surface-active compounds such as sodium lauryl sulfate (SLS; sodium dodecyl sulfate; SDS), poloxamer 407, and cocamidopropyl betaine. Zinc ions and L-arginine have also been added to toothpaste. These compounds reduce the amount of extracellular polymeric substances within the biofilm, making it easier to remove dental plaque by shear forces [21].

Since the introduction of fluoride in drinking water and toothpaste, there has been a decline in the prevalence of caries [22,23]. The fluoride content in mouthwashes is usually 225 ppm (equivalent to 0.0497% sodium fluoride), while it is much higher in toothpaste (e.g., 1450 ppm), where a combination of different fluoride compounds such as sodium fluoride, stannous fluoride, and amine fluoride is frequently used [20]. Fluoride has a dual function [22]. It strengthens the teeth by forming fluoride hydroxyapatite that is deposited on the enamel surface [22]. Fluoride hydroxyapatite is more resistant to degradation by acids than hydroxyapatite [22]. The release of hydroxide in exchange with fluoride leads to an increase in the pH value of the microenvironment. In addition, fluoride has an anti-bacterial effect, albeit with a relatively high minimum inhibitory concentration (MIC) of about 600 μg/mL [24]. Fluoride inhibits the F_1_F_0_-ATPase complex, which is important for proton efflux and glucan synthesis [22]. However, excessive exposure to fluoride can lead to fluorosis [25].

CHX is a cationic bisbiguanide compound with broad-spectrum antiseptic properties against Gram-positive bacteria, Gram-negative bacteria, fungi, and viruses [26,27]. CHX has the advantage of being able to attach to negative charges of tooth biofilms through one of its cationic moieties, keeping the other cationic moiety free to interact with bacteria [28]. This leads to high substantiveness of the compound that may remain active in the oral cavity for up to 12 h [28]. The anti-bacterial action of CHX relies on its interaction with negative-charged sites in the cell wall and bacterial membranes, resulting in membrane rupture and cytoplasmic leakage [27,29]. The two cationic moieties of CHX may also lead to the cross-linking of negatively charged phospholipids in the bacterial membrane, thereby replacing Ca^2+^ ions from the membrane and resulting in its destabilization [27,30]. Moreover, CHX induces the production of reactive oxygen species (ROS) and causes lipid peroxidation of unsaturated fatty acids, resulting in biochemical alterations in membrane proteins, phospholipids, carbohydrates, and nucleic acids [31]. CHX also inhibits glycosyltransferase and the phosphoenolpyruvate phosphotransferase system (PTS), thereby perturbing glucan production, sugar membrane transport, and glycolytic metabolism [29]. CHX has several adverse effects including tooth staining, taste disturbance, and burning, which are usually reversible [32]. Other limitations of CHX may be the acquisition of resistance mechanisms [27,33], allergic reactions, and severe anaphylactic shock in rare cases [34].

CPC is a monocationic quaternary ammonium compound with a hydrophobic hexadecane chain that has become more popular than CHX in mouthwashes because of its low toxicity and because it lacks several of the adverse effects of CHX [32]. This is despite the lower anti-bacterial potency of CPC compared with CHX [18]. The bactericidal action mechanism of CPC involves binding to negative charges in the cell wall and membrane and integration of its hydrophobic tail into the phospholipid layer, resulting in membrane disorganization and eventual bacterial lysis [30,35].

Triclosan is a non-ionic aromatic disinfectant with a broad anti-bacterial spectrum. It integrates into the lipid bilayer resulting in bacterial lysis [36]. Triclosan can also alter the 3D structure of proteins including superoxide dismutase (SOD), enoyl-acyl carrier protein reductase (FabI) involved in fatty acid synthesis, and various viral proteins, thereby interfering with their activities [36,37,38,39]. It induces the production of ROS, among others, by its ability to conduct electron transfer [40,41]. The increased ROS production by triclosan can lead to the upregulation of beta-lactamase and multidrug efflux pumps involved in antibiotic resistance [40].

Several reports have shown an anti-bacterial and anti-biofilm activity of polyunsaturated fatty acids such as ω-3 (e.g., α-linolenic acid and docosahexaenoic acid) and ω-6 (e.g., arachidonic acid (AA)) against various Gram-positive bacteria including *S. mutans*, *Streptococcus pneumoniae*, and *Staphylococcus aureus* [42,43,44,45,46,47,48]. AA was found to induce lipid peroxidation in *Staph. aureus* [43]. The anti-bacterial activity of AA was attenuated by catalase and α-tocopherol (vitamin E), while it was enhanced by ascorbic acid (vitamin C), suggesting the involvement of lipid peroxidation and hydrogen peroxide (H_2_O_2_) in the killing of *Staph. aureus* [42,43]. Moreover, AA is believed to be oxidized to reactive electrophiles that modify macromolecules, resulting in a bactericidal effect [43]. Another anti-bacterial mechanism of AA is the alteration of fatty acid homeostasis with a reduced expression of genes involved in fatty acid synthesis [46]. AA was found to increase the susceptibility of *Staph. aureus* to aminoglycosides [49] and monounsaturated palmitoleic acid increased the susceptibility of *Staph. aureus* to vancomycin [50]. Our recent finding that AA has anti-bacterial and anti-biofilm effects against *S. mutans* [44] encouraged us to investigate the feasibility of using this natural fatty acid in mouthwashes together with commonly used antiseptics. Here, we present data showing an antagonistic effect of AA with the cationic CHX or CPC at certain concentrations, while the combined treatment of AA with non-ionic triclosan had an additive effect on *S. mutans*. The addition of fluoride to the AA/triclosan combination further increased the anti-bacterial and anti-biofilm activity against *S. mutans*. The cariogenic *S. mutans* strain UA159 is a clinical isolate that was used in this study as the model bacterium.

## 2. Results

### 2.1. Arachidonic Acid Increased the Anti-Bacterial Effect of Triclosan on Streptococcus mutans

We recently observed that the naturally occurring polyunsaturated fatty acid arachidonic acid (AA) has anti-bacterial and anti-biofilm activities against the cariogenic *Streptococcus mutans* UA159 [44]. In order to understand the applicability of AA in dentistry, it was important to investigate how AA affects the anti-bacterial activity of antiseptics commonly used in mouthwashes and other dentifrices. To this end, we exposed the bacteria to various combinations of AA (1.56–25 μg/mL) and one of the following three antiseptics: chlorhexidine (CHX) (0.078–5 μg/mL), cetylpyridinium chloride (CPC) (0.078–5 μg/mL), or triclosan (0.3125–20 μg/mL), for 24 h using the checkerboard assay. Among the antiseptics applied alone, CHX was the most potent compound with an MIC of 0.625 μg/mL, followed by CPC with an MIC of 5 μg/mL, while triclosan was the least potent, with an MIC of 20 μg/mL (Figure 1A–C). The MIC of AA was between 12.5 and 25 μg/mL (Figure 1D). AA did not augment the anti-bacterial effect of CHX or CPC, and even at certain concentrations, there was an antagonistic effect (Figure 2A,B). The antagonism was in particular observed in the combination of 0.3125 or 0.625 μg/mL CHX with 25 μg/mL AA (Figure 2A), 2.5 μg/mL CPC with 3.125–12.5 μg/mL AA (Figure 2B), and 5 μg/mL CPC with 25 μg/mL AA (Figure 2B). On the other hand, AA had an additive effect on triclosan (Figure 2C). The additive effect was evident when combining 5 μg/mL triclosan with 6.25 or 12.5 μg/mL AA and when combining 10 μg/mL triclosan with 3.125, 6.25, or 12.5 μg/mL AA (Figure 2C; FICI of 0.625–0.75). No bacterial regrowth was observed with these combinations even after a 48 h incubation.

### 2.2. Fluoride Increased the Anti-Bacterial Effect of Arachidonic Acid and Triclosan on Streptococcus mutans

Since sodium fluoride (NaF) is commonly used in dental medications, we also investigated the triple combinations of AA, triclosan, and NaF at different concentrations in a modified checkerboard assay. NaF alone showed an MIC of 500 μg/mL (Figure 1E) and increased the anti-bacterial activity of AA (Figure 3A; FICI of 0.625–0.75) and triclosan (Figure 3B; FICI of 0.625–0.75). The additive effect of NaF was observed in the combination of 6.25 μg/mL AA with 7.8–125 μg/mL NaF (Figure 3A) and when 5 μg/mL triclosan was combined with 15.6–250 μg/mL NaF (Figure 3B). Since the additive effect of NaF on the anti-bacterial activity of AA was already observed at 7.8 μg/mL and reached a maximum at 31.2 μg/mL (Figure 3A), we decided to examine the 15 μg/mL and 30 μg/mL concentrations of NaF in the triple checkerboard assay (Figure 3C). Based on the finding that 5 μg/mL triclosan had a significant additive anti-bacterial effect in combination with AA (Figure 2C), we chose to use the lower concentrations of 1.25 and 2.5 μg/mL triclosan for the triple combination. The addition of NaF to AA/triclosan did not result in a significant further increase in anti-bacterial activity for most of the combinations, with the exception of 30 μg/mL NaF, which enhanced the anti-bacterial effect of 6.25 μg/mL AA with 2.5 μg/mL triclosan (Figure 3C). Importantly, there was no antagonistic effect between these three compounds (Figure 3C), making this combination suitable as a treatment module.

### 2.3. The Combined Treatment of Arachidonic Acid and Triclosan Increased Membrane Perforation

We next examined the effect of the three compounds, alone or in combination, on membrane perforation after a 2 h incubation, which was determined by the uptake of propidium iodide (PI). As expected, increased PI uptake was observed with increasing concentrations of AA, reaching 76 ± 8% at 12.5 μg/mL AA (Figure 4A–D,U), which is consistent with the report by Chamlagain et al. [44]. Triclosan had only a weak effect on PI uptake, reaching 20 ± 1% at 10 μg/mL (Figure 4E–H,U), which is consistent with the study by Avraham et al. [51]. In contrast, NaF did not increase PI uptake after a 2 h incubation, even at the highest tested dose of 240 μg/mL (Figure 4I–L,U). NaF at a dose of 30 μg/mL did not significantly alter the PI-uptake caused by triclosan (Figure 4M,N,U), but enhanced the PI-uptake caused by 6.25 μg/mL AA (47 ± 5% vs. 39 ± 4% in the absence of NaF; Figure 4P vs. Figure 4C and Figure 4U). Triclosan enhanced the PI uptake caused by AA, with 35 ± 3% PI-positive cells when *S. mutans* was treated with 3.125 μg/mL AA and 5 μg/mL triclosan (Figure 4Q,U) and 71 ± 6% PI-positive cells when the bacteria were treated with 6.25 μg/mL AA and 2.5 μg/mL triclosan (Figure 4S,U). Surprisingly, the addition of 30 μg/mL NaF to the combined AA/triclosan treatment resulted in decreased PI uptake (Figure 4R,T vs. Figure 4Q,S and Figure 4U), suggesting that NaF prevents PI uptake or somehow affects PI staining despite decreased viability, as shown in Figure 3C. Other researchers have also observed low PI uptake of NaF-treated *S. mutans* despite reduced viability and biofilm formation [24,52].

### 2.4. Arachidonic Acid, but Not Triclosan and Fluoride, induced Significant Membrane Hyperpolarization of Streptococcus mutans

Bacteria must maintain a proton motive force (PMF) for performing vital biochemical processes [53,54]. It was therefore important to investigate the effect of the three compounds alone or in combination on the bacterial membrane potential. For this purpose, we used the potentiometric dye DiOC2(3) after a 10 min incubation of the bacteria with the different combinations of compounds and control bacteria. A shift in the fluorescence emission wavelength from green to red indicates membrane hyperpolarization. As expected [44], AA at a dose of 12.5 μg/mL resulted in membrane hyperpolarization (Figure 5A,B,I). Triclosan induced a minor hyperpolarization at 10 μg/mL (Figure 5C,D,I), while NaF showed no significant effect on the membrane potential (Figure 5E,F,I). Triclosan and NaF had no significant effect on the AA-induced hyperpolarization in either double or triple treatment (Figure 5G–I).

### 2.5. The Triple Treatment Had a Better Anti-Biofilm Effect Than Each Agent Alone

Based on the promising effect of the AA/triclosan/NaF triple treatment on planktonic growth (Section 2.1 and Section 2.2), we were prompted to investigate the combined treatment on biofilm formation, which is a major virulence factor of *S. mutans* [55]. An increased anti-biofilm effect was observed when NaF was combined with AA (Figure 6A and Figure 7A) or triclosan (Figure 6B and Figure 7B). NaF by itself showed a dose-dependent reduction in biofilm metabolic activity with an MBIC of 250–500 μg/mL (Figure 6A,B). Further reduced metabolic biofilm activity was observed when 6.25 μg/mL AA was combined with 31.25–125 μg/mL NaF (Figure 6A) and when 5 μg/mL triclosan was combined with 15.6–250 μg/mL NaF (Figure 6B). The biofilm biomass, which also includes the extracellular matrix enwrapping the bacteria, was significantly diminished by 6.25 μg/mL AA together with 62.5–125 μg/mL NaF (Figure 7A) and by 5 μg/mL triclosan together with 15.6–250 μg/mL NaF (Figure 7A). NaF at 15 and 30 μg/mL enhanced the anti-biofilm effect of 3.125 μg/mL AA in combination with 2.5 μg/mL triclosan (Figure 6C and Figure 7C).

### 2.6. Confocal Microscopy of Triple-Treated Biofilms in Comparison to Single and Double Treatments

To further prove the enhanced anti-biofilm effect of triple treatment, we performed confocal microscopy of biofilms formed after a 24 h incubation with various combinations of AA, triclosan, and fluoride at sub-minimum biofilm inhibitory concentrations (MBICs). Biofilms generated under similar conditions in the absence of agents or equal ethanol concentrations served as controls. The biofilms were stained with the following dyes: 1. SYTO 9, which enters both live and dead bacteria, emits green fluorescence when bound to nucleic acid; 2. propidium iodide (PI), which is positively charged and can only enter bacteria when the membrane is perforated, emits red fluorescence when bound to nucleic acids; and 3. AlexaFluor^647^-conjugated Dextran 10,000, which is incorporated into the extracellular biofilm matrix during biofilm formation, emits far-red fluorescence. Fluorescent Dextran is presented in the images in blue color. When used individually, 3.125 μg/mL AA and 2.5 μg/mL triclosan slightly reduced the number of bacteria in the biofilms by 15–20% (Figure 8 and Figure 9A,D), and both compounds increased the proportion of PI-positive cells (23 ± 7% increase for AA and 71 ± 10% increase for triclosan; Figure 8 and Figure 9B,D). In addition, 3.125 μg/mL AA reduced EPS staining by 35–45%, while no significant effect was observed with 2.5 μg/mL triclosan (Figure 8 and Figure 9C,D). Biofilms generated in the presence of 30 μg/mL NaF showed similar intensity of SYTO 9 staining (Figure 8 and Figure 9A,D), while a significant reduction in the PI-staining (89 ± 1% reduction) was observed (Figure 8 and Figure 9B,D). NaF at 30 μg/mL increased EPS staining by 18 ± 10% as a single agent (Figure 8 and Figure 9C,D) and enhanced the anti-biofilm effect of 3.125 μg/mL AA and 2.5 μg/mL triclosan, as shown by reduced SYTO 9 staining (Figure 8 and Figure 9A,D). Surprisingly, NaF also significantly reduced PI staining when combined with AA or triclosan (Figure 8 and Figure 9B,D), while it increased EPS staining (Figure 8 and Figure 9C,D). The combined treatment of 3.125 μg/mL AA with 2.5 μg/mL triclosan had stronger anti-biofilm activity than either agent alone, which was further enhanced by the addition of 30 μg/mL NaF (Figure 8 and Figure 9B,D).

### 2.7. HR-SEM Images of Triple-Treated Biofilms

It was also important to look at the morphology of the biofilms formed in the presence of the compounds. The panoramic images at ×1000 magnification in Figure 10 show the biofilm coverage and the gross topography, while the images at ×5000 and ×20,000 magnifications in Figure 11 and Figure 12 are focused on areas with bacteria to visualize the bacterial morphology and the presence of EPS, which appears as a greyish diffuse mass covering the bacteria. In the panoramic images, we could see full coverage by the control biofilms, which had several bacterial clusters (Figure 10A). Bacterial clusters could also be observed in biofilms formed in the presence of 30 μg/mL NaF, 2.5 μg/mL triclosan, and the combined NaF/triclosan treatment, although the coverage was reduced with NaF alone and the combined NaF/triclosan treatment (Figure 10C,D,G). AA at 3.125 μg/mL showed scattered bacteria with few bacterial clusters (Figure 10B), and the coverage was further reduced in the presence of NaF and/or triclosan with some bacterial clusters (Figure 10E,F,H). At the higher magnification, the classical structure of *S. mutans* biofilms could be seen in the control samples where bacteria in multilayers form circular mountain ridges with EPS-covered valleys (Figure 11A and Figure 12A). AA treatment led to single to double-cell layers of bacteria held in chains with around 50% coverage and almost no EPS (Figure 11B and Figure 12B). NaF-, triclosan-, and NaF/triclosan-treated biofilms were multilayered with clear EPS content, which was more pronounced in the double-treated samples (Figure 11C,D,G and Figure 12C,D,G). The bacterial organization in the triclosan-treated biofilms differed from the control bacteria with a more flattened structure at the ridges and less deep valleys (Figure 11C and Figure 12C vs. Figure 11A and Figure 12A). Incubating the bacteria with both NaF and triclosan caused a reorganization of the biofilms with clusters of bacteria enwrapped in EPS (Figure 11G and Figure 12G). The biofilms of the combined AA/NaF treatment resembled that of the AA mono-treatment (Figure 11E vs. Figure 11B and Figure 12E vs. Figure 12B). Higher magnification of the few bacterial clusters that attached to the glass pieces after AA/triclosan or AA/triclosan/NaF treatment showed dysmorphic, crumpled bacteria with distorted membranes and surrounded by cell debris (Figure 11F,H and Figure 12F,H), suggesting that the bacteria were dead.

## 3. Discussion

The main objective of this study was to investigate the applicability of arachidonic acid (AA) as an anti-caries agent in combination with antiseptics commonly used in mouth rinses and dentifrices. AA is a natural unsaturated Ω-6 fatty acid (20:4) found in animal products such as poultry, fish, meat, and eggs [56]. In the human body, it is covalently bound to phospholipids in cell membranes, where it influences membrane fluidity and the activity of certain enzymes and ion channels and modulates synaptic activity in the brain [56]. AA, which is released by macrophages when they are stimulated [57,58], is thought to help fight bacteria because of its anti-bacterial action [47] in addition to increasing the phagocytic activity of macrophages and neutrophils and their ability to kill engulfed bacteria [46,47,59,60]. In addition, it stimulates NADPH oxidase activity in neutrophils, resulting in increased ROS production [61], which is involved in the killing of bacteria by these innate immune cells [62]. Since AA is oxidized into radicals by ROS with consequent lipid peroxidation, an increase in ROS production would enhance the anti-bacterial effect of AA [43]. AA can be converted into both inflammatory mediators, such as prostaglandins, and anti-inflammatory mediators, such as the endocannabinoids arachidonoylethanolamine (anandamide) and 2-arachidonoylglycerol (2-AG), which have a short half-life and act locally to dampen immune responses [56]. The human body has evolved mechanisms to keep free AA levels low through the Lands reacylation/deacylation cycle [63]; therefore, any use of AA is likely to be limited to local effects, which is an advantage for oral therapy. Importantly, AA has low cytotoxicity in both in vitro toxicity assays [44] and in mice and hamsters [64]. Besides having anti-bacterial activity [43,44,46], AA is able to kill *Schistosoma* parasites in the larval, juvenile, and adult stages and reduce the worm burden of *Schistosoma*-infected mice and hamsters [64]. In combination with praziquantel, AA resulted in a high cure rate in both children with mild and severe *Schistosoma* infections [64,65]. These data indicate that AA is a potential anti-microbial and anti-parasitic drug that can be used for clinical purposes.

The three antiseptics, i.e., CHX, CPC, and triclosan, used in this study belong to three different chemical classes and show different anti-bacterial potency against *S. mutans*, with CHX being the most potent, followed by CPC, and the least potent is triclosan (Figure 1). This goes along with other studies showing the same order of potency [18,66,67,68,69,70]. Notably, there was an antagonistic effect between the two antiseptic compounds CHX and CPC and AA at certain concentrations (Figure 2A,B). CHX has been shown to interact with acidic groups of glycoproteins, an action that has been associated with reduced plaque adhesion [71,72]. It is likely that the antagonistic effect observed between CHX and AA at certain concentrations is caused by a chemical interaction between CHX and AA that prevents the compounds from reaching their bacterial targets. CHX has ten nitrogen atoms, two of which are positively charged [73]. These cations can interact with the negatively charged carboxyl group of AA, and the short aliphatic hexamethylene between two (p-chlorophenyl)biguanide moieties of CHX can theoretically interact with the aliphatic chain of AA through hydrophobic forces. Previous studies have observed an antagonistic effect of the negatively charged fluoride on CHX substantivity [74] and an inhibition of the anti-bacterial effect of CHX by anionic lubricating gel formulations [75] and the anionic surfactant SLS [76]. Since the anti-bacterial action of CHX relies on its binding to negatively charged phospholipids in the bacterial membrane, leading to its perforation [73], any neutralization of its positive charges would reduce its anti-bacterial activity.

CPC has one positively charged nitrogen, which may interact with the carboxyl group of AA. In addition, CPC has a long aliphatic tail (16 carbons), which may compete with the unsaturated aliphatic chain of AA (20 carbons) for membrane intercalation sites, or, alternatively, they may interact with each other by hydrophobic forces, resulting in the neutralization of their anti-bacterial action. Such an interaction usually occurs at a certain molar ratio, which can explain why the antagonistic effect was only observed at certain concentrations. The study of Evans et al. [76] showed that NaF had an antagonistic effect on the anti-bacterial activity of CPC against *S. mutans*.

Unlike CHX and CPC, an additive anti-bacterial effect was observed between AA and triclosan (Figure 2C), which is a neutral molecule without an aliphatic moiety. The observed additive effect suggests that the two compounds have different complementary targets. Triclosan is known to inhibit the enoyl-acyl reductase (ENR) transporter protein (FabI) involved in type II fatty acid synthesis [77] and membrane lipid biosynthesis [78]. Type II fatty acid synthesis (FASII) is unique to bacteria and essential for their survival [79,80,81]. Some researchers have argued that exogenously added fatty acids and serum-containing fatty acids can antagonize the effect of FASII inhibitors by circumventing the need for endogenous fatty acid synthesis [82]. However, the addition of AA to triclosan increased anti-bacterial activity towards *S. mutans* (Figure 2C), thereby ruling out the above compensatory hypothesis in this case. One mechanism for the additive effect could be the inhibition of fatty acid synthesis by AA, as shown by Eijkelkamp et al. [46] for *S. pneumoniae*, where it reduced the expression of genes involved in fatty acid biosynthesis and perturbed metabolic processes such as carbon-source utilization. Moreover, *S. mutans* treated with AA showed a downregulation of the genes *fabM* and *fabD* involved in fatty acid homeostasis [44]. The effect of AA on fatty acid synthesis is only one of its many anti-bacterial action mechanisms. As mentioned above, AA incorporated into the bacterial membrane reacts with ROS produced in the bacteria, resulting in the production of radicals and lipid peroxidation, which is detrimental to bacteria [43]. The increased ROS production induced by triclosan [83] may further contribute to the increased anti-bacterial effect observed in combination with AA (Figure 2C).

Since many dental products contain fluoride, it was also important to investigate the compatibility of fluoride with AA and triclosan. A study by Mellberg et al. [84] showed that triclosan does not interfere with fluoride-mediated remineralization, which is essential for the caries-inhibiting effect of fluoride. We observed here that fluoride enhanced the anti-bacterial effect of both AA (Figure 3A) and triclosan (Figure 3B), with improved anti-bacterial activity of the triple treatment (Figure 3C). Of the three compounds, AA had the strongest effect on membrane perforation and hyperpolarization as determined by PI uptake and a green-to-red shift in the DiOC2(3) fluorescence (Figure 4 and Figure 5). The addition of triclosan to AA increased the percentage of bacteria with membrane perforation (Figure 4) without enhancing membrane hyperpolarization (Figure 5). Remarkably, the presence of fluoride led to reduced PI staining of bacteria treated with AA and/or triclosan (Figure 4), despite increased anti-bacterial activity (Figure 3C). This was quite surprising. A similar low PI staining of fluoride-treated *S. mutans* biofilms despite reduced viability has also been observed by some other researchers [24,52]. The reason behind this phenomenon is unclear. It could be that the negatively charged fluoride interacts with the positively charged PI through electrostatic forces, impairing its ability to bind nucleic acids. Fluoride has also been shown to bind DNA, which is accompanied by DNA damage [85] and DNA unfolding [86]. The binding of fluoride to DNA may interfere with the ability of PI to intercalate into these structures. A study by Samanta et al. [87] showed that PI interacts with anionic surfactants such as SLS, resulting in fluorescence quenching [87]. It is likely that a similar quenching mechanism is involved with fluoride, which is the most electronegative element.

An important feature of AA, triclosan, and fluoride is their ability to reduce biofilm formation of *S. mutans* as single agents and to further reduce biofilm formation when used in combination (Figure 6 and Figure 7). The respective minimum biofilm inhibitory concentration (MBIC) was 12.5 µg/mL for AA, 10 µg/mL for triclosan, and 500 µg/mL for NaF (Figure 6). For most combinations, anti-biofilm activity goes along with anti-bacterial activity (Figure 6 vs. Figure 3). However, some combinations caused a weaker or stronger anti-biofilm effect (Figure 6 vs. Figure 3). For instance, the combination of fluoride with triclosan required higher doses of the two compounds to achieve an anti-biofilm effect (Figure 6B vs. Figure 3B). This was especially seen for 125–250 µg/mL NaF in combination with 0.3125–1.25 µg/mL triclosan (Figure 6B vs. Figure 3B). The reason for this phenomenon could be the stimulation of biofilm formation by triclosan and NaF at sub-MBIC concentrations through the induction of biofilm-associated genes [88,89]. This was also observed by confocal microscopy (Figure 8 and Figure 9) and HR-SEM (Figure 11 and Figure 12), where higher amounts of exopolysaccharides were present in samples treated with NaF and triclosan. However, the addition of 3.125–6.25 µg/mL AA to the combined triclosan/NaF treatment resulted in a stronger anti-biofilm effect than anti-bacterial activity (Figure 6C vs. Figure 3C), indicating a specific anti-biofilm activity. This was also demonstrated by confocal microscopy (Figure 8 and Figure 9) and HR-SEM (Figure 10, Figure 11 and Figure 12). Similar to the flow cytometry data of planktonic bacteria stained with SYTO 9/PI (Figure 4), the PI staining of fluoride-treated biofilms was significantly reduced (Figure 8 and Figure 9), despite the further reduced biofilm formation when combined with AA and/or triclosan (Figure 8 and Figure 9) and the appearance of dysmorphic bacteria on HR-SEM images (Figure 11 and Figure 12). The reduced PI staining is apparently not due to the presence of live bacteria with intact membranes, but rather due to the interference of fluoride with PI, as discussed above. Three different surfaces were used for biofilm formation as follows: tissue culture-grade polystyrene wells with a negatively charged hydrophilic surface, ibiTreat slides with a slightly negatively charged hydrophilic surface, and uncharged glass pieces. Although the structure of the biofilm is influenced by the surface properties, the increased anti-biofilm effect of the triple treatment was observed regardless of the type of surface.

Altogether, our data show that the combination of AA with triclosan and fluoride enhances the anti-bacterial and anti-biofilm effects compared with the single and double treatments. Moreover, AA prevents the enhanced biofilm formation caused by sub-MBIC concentrations of triclosan and fluoride. The data obtained in this study suggest a potential beneficial contribution of AA in improving the efficacy of anti-cariogenic triclosan/fluoride treatment. A limitation of this study is that it is an in vitro study using only one clinically relevant cariogenic bacterium. Future studies should focus on in vivo effects and how the triple combination affects the oral microbiota and caries formation.

## 4. Materials and Methods

### 4.1. Bacteria and Cultivation

The cariogenic *Streptococcus mutans* UA159 strain (ATCC 700610) isolated from a child with caries, was used as the cariogenic bacterial model strain in our study. It was cultured in brain heart infusion (BHI) broth (HiMedia Laboratories Pvt. Ltd., Maharashtra, India), for planktonic-growing conditions, and in BHI supplemented with 2% sucrose, for biofilm-producing conditions. A starter culture was prepared the day before the experiment by inoculating 100 μL of a frozen stock into 10 mL of BHI, followed by overnight incubation at 37 °C in a humidified incubator with 5% CO_2_ [44]. All experiments were performed under these incubation conditions.

### 4.2. Viability Assay

To study the effect of various combinations of chlorhexidine gluconate (CHX, Sigma, St. Louis, MO, USA), cetylpyridinium chloride (CPC, Sigma), sodium fluoride (NaF, Sigma), and arachidonic acid (AA, >99% purity, Nu-Check Prep, 109 W Main St, Elysian, MN, USA), an overnight starter culture of *S. mutans* UA159 was diluted to an initial optical density (OD_600nm_) of 0.1 in BHI, and 100 μL of this bacterial suspension was added to 100 μL of the various agents ×2 concentration in flat-bottomed 96-tissue-grade well plates (Corning Incorporation, Kennebunk, ME, USA) resulting in an initial bacterial OD_600nm_ of 0.05 and ×1 concentration of the agents [44]. For checkerboard assays of combined treatment, stock solutions of different agent concentrations were prepared ahead and used for the intended wells in horizontal vs. vertical directions [51]. Initially, single agents were used to define the minimum inhibitory concentration (MIC) for the UA159 strain. Then, double treatments were performed for each combination (AA/CHX, AA/CPC, AA/triclosan, AA/NaF, and triclosan/NaF). After determining the MIC of the combined treatment, a triple checkerboard assay was performed with various combinations of AA, triclosan, and NaF. Triplicates were performed for each experiment, and each double and triple checkerboard assay was performed three times to verify the data. Control samples were bacteria-incubated in BHI only and ethanol at 0.1%, which was the highest ethanol concentration used in the combined treatments. The samples were incubated for 24 h in a humidified incubator at 37 °C with 5% CO_2_. At the end of incubation, the turbidity at 600 nm, which reflects the amount of bacteria, was measured in a Multiskan SkyHigh microplate reader (ThermoScientific, Life Technologies Holdings Pte Ltd., Singapore). Background reads were the agents in BHI without bacteria. The % turbidity was calculated according to the following equation:$$\%\;\mathrm{Turbidity}=\frac{{OD}_{600\;nm}\;of\;treated\;samples-{OD}_{600\;nm}\;of\;background}{{OD}_{600\;nm}\;of\;control\;samples-{OD}_{600\;nm}\;of\;background}\times 100\%$$

MIC was determined by the concentration resulting in no visible bacterial growth. The fractional inhibitory concentration index (FICI) was calculated to determine whether the combined treatment was synergistic, additive, or antagonistic [51,90].

### 4.3. SYTO 9/PI Live/Dead Staining by Flow Cytometry

*S. mutans* at an initial OD_600nm_ of 0.3 was exposed to different combinations of compounds in 1 mL BHI for 2 h. At the end of incubation, the samples were centrifuged at 5000× *g* for 5 min, and the bacterial pellet was resuspended in PBS containing 3.3 μM SYTO 9 (Molecular Probes, Life Technologies, Carlsbad, CA, USA) and 10 μg/mL propidium iodide (PI) (Sigma, St. Louis, MO, USA). After a 20 min incubation at room temperature, the fluorescence intensities were measured in an LSR-Fortezza flow cytometer (BD Biosciences, San Jose, CA, USA) using the excitation/emission of 488 nm/520 nm for SYTO 9 and 561 nm/586 nm for PI [44]. A total of 50,000 events were collected for each sample using BD FACSDiva software 8.0.1, and each treatment group was performed in triplicates. The analysis was performed using De Novo FCS Express 7.12.0007 software. Increased PI uptake is an indication of dying bacteria with perforated membranes.

### 4.4. Membrane Potential Determination by Flow Cytometry

To determine the effect on membrane potential, *S. mutans* exposed to various combinations of the compounds in PBS for 10 min, were stained with 30 μM of the potentiometric dye DiOC2(3) for an additional 20 min (BacLight Membrane Potential Kit, Molecular Probes, Life Technologies, Eugene, OR, USA). The green and red fluorescence values were measured in a Fortezza flow cytometer using the respective excitation/emission of 488 nm/530 nm and 488 nm/620 nm. A total of 50,000 events were collected for each sample, and each treatment group was performed in triplicates [44]. A relative increase in red fluorescence in comparison with green fluorescence is an indication of membrane hyperpolarization.

### 4.5. Biofilm Assay

The ability of various combinations of test agents to affect biofilm formation was performed similarly to the viability assay described in Section 4.2, except that the bacteria were incubated in BHI supplemented with 2% sucrose. After a 24 h incubation under static conditions, the biofilms were washed twice with 200 μL PBS and then either exposed to 50 μL of 0.5 mg/mL MTT (Sigma) in PBS for 1 h at 37 °C to measure the metabolic activity or 100 μL of a 0.25% crystal violet (CV) solution (Merck KGaH, Darmstadt, Germany) for 20 min at room temperature to measure the biofilm biomass [44]. At the end of incubation, 180 μL PBS was added to the MTT-exposed biofilms, and the supernatant was decanted. The tetrazolium formed was dissolved in 200 μL dimethyl sulfoxide (DMSO), and the absorbance at 570 nm was measured in a Multiskan SkyHigh microplate reader. The CV-stained biofilms were washed several times with ddw to remove excess stain, the CV staining of biofilms was dissolved in 200 μL of 33% acetic acid, and the absorbance at 595 nm was measured in a Multiskan SkyHigh microplate reader. Triplicates were performed for each experiment, and each double and triple checkerboard assay was performed three times to verify the data. The percentage of metabolic active bacteria in biofilms and the biofilm biomass of the treated samples was calculated in comparison to control biofilms after subtracting the background. The minimum biofilm inhibitory concentration (MBIC) was defined as the lowest concentration that resulted in no visible biofilm formation.

### 4.6. Confocal Microscopy of Biofilms

The amount of live/dead bacteria in 24-h-treated biofilms formed on ibiTreat 8-well μ-slides (ibidi GmbH, Gräfelfing, Germany) was determined by staining the biofilms with 3.3 μM SYTO 9 and 10 μg/mL PI in PBS for 20 min [51]. The presence of exopolysaccharides (EPSs) was detected by adding 5 μg/mL Alexafluor^647^-conjugated Dextran 10,000 (Molecular Probes Inc., Eugene, OR, USA) during biofilm formation. The stained biofilms were washed with PBS, fixed with 4% paraformaldehyde (Electron Microscopy Sciences, Hatfield, PA, USA) in PBS, and mounted with 50% glycerol prior to visualization using a scanning disk confocal microscope (Nikon Corporation, Tokyo, Japan). Images were captured at 2.5 μm intervals, using the excitation laser 488 nm for SYTO 9 (green fluorescence), the excitation laser 561 nm for PI (red fluorescence), and the excitation laser 640 nm for AlexaFluor^647^-Dextran 10,000 (far-red fluorescence, presented in the images in blue). NIS elements Ar (Advanced Research) software version 5.21.03 (Nikon Instruments Inc.) was used to prepare 3D images and calculate the fluorescence intensities in each layer of each biofilm. A total of 7–10 images were captured for each treatment taken from 3–4 different replicates.

### 4.7. High-Resolution Scanning Electron Microscopy (HR-SEM) of Biofilms

Biofilms formed after a 24 h incubation on glass pieces were washed in ddw, fixed with 4% glutaraldehyde (Electron Microscopy Sciences, Hatfield, PA, USA) in ddw for 2 h, washed again in ddw, and dried. The samples were then sputtered with iridium and visualized by an analytical high-resolution scanning electron microscope (HR-SEM) (Apreo 2S LoVac, ThermoScientific) at various magnifications.

### 4.8. Statistical Analysis

For each experiment performed in triplicates, the statistical significance was calculated by comparing two or more treatments using one-way ANOVA with ad hoc corrections. A *p*-value below 0.05 was considered statistically significant.

## 5. Conclusions

We investigated the anti-bacterial and anti-biofilm effects of various combinations of arachidonic acid (AA), chlorhexidine, cetylpyridinium chloride, triclosan, and fluoride against the cariogenic bacterium *Streptococcus mutans*. While an antagonistic effect was observed between AA and chlorhexidine or cetylpyridinium chloride at certain concentrations, an additive effect was observed between AA and triclosan. The addition of fluoride to the combined AA/triclosan treatment further enhanced the anti-bacterial and anti-biofilm effects. These data suggest that AA is a potential anti-bacterial and anti-biofilm drug for *S. mutans* that can be added together with triclosan and fluoride to mouth rinses and other dentifrices to improve oral health.

## Figures and Tables

**Figure 1 antibiotics-13-00540-f001:**
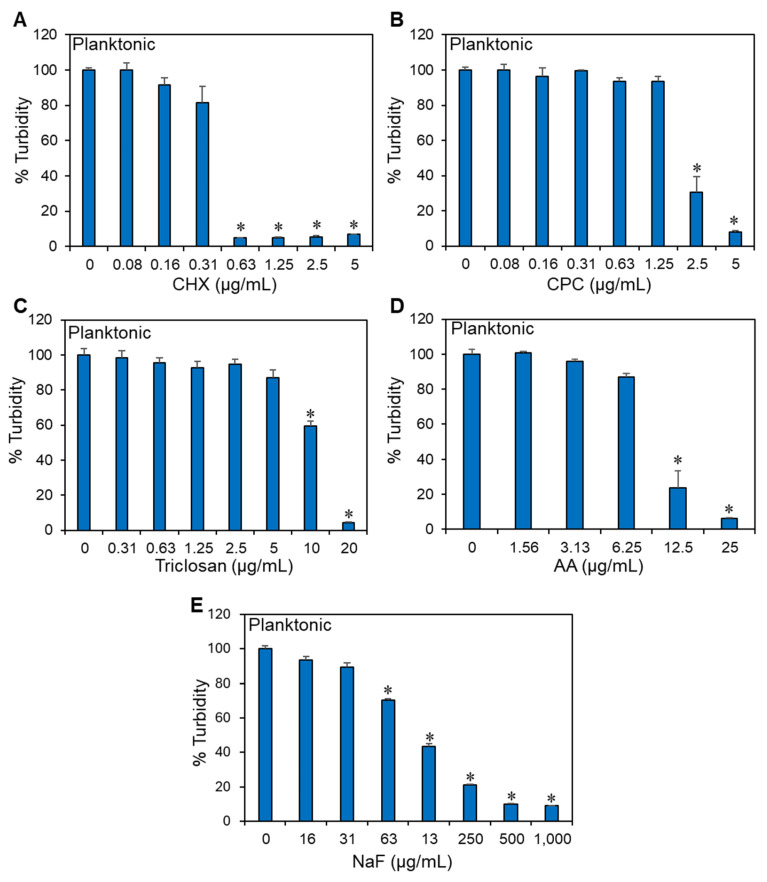
Anti-bacterial activities of the different compounds used in this study against *S. mutans*. (**A**–**E**). The percentage turbidity of planktonic-growing *S. mutans* after a 24 h incubation with chlorhexidine (CHX; (**A**)), cetylpyridinium chloride (CPC; (**B**)), triclosan (**C**), arachidonic acid (AA; (**D**)), or sodium fluoride (NaF; (**E**)). n = 3. * *p* < 0.05 when compared to control samples.

**Figure 2 antibiotics-13-00540-f002:**
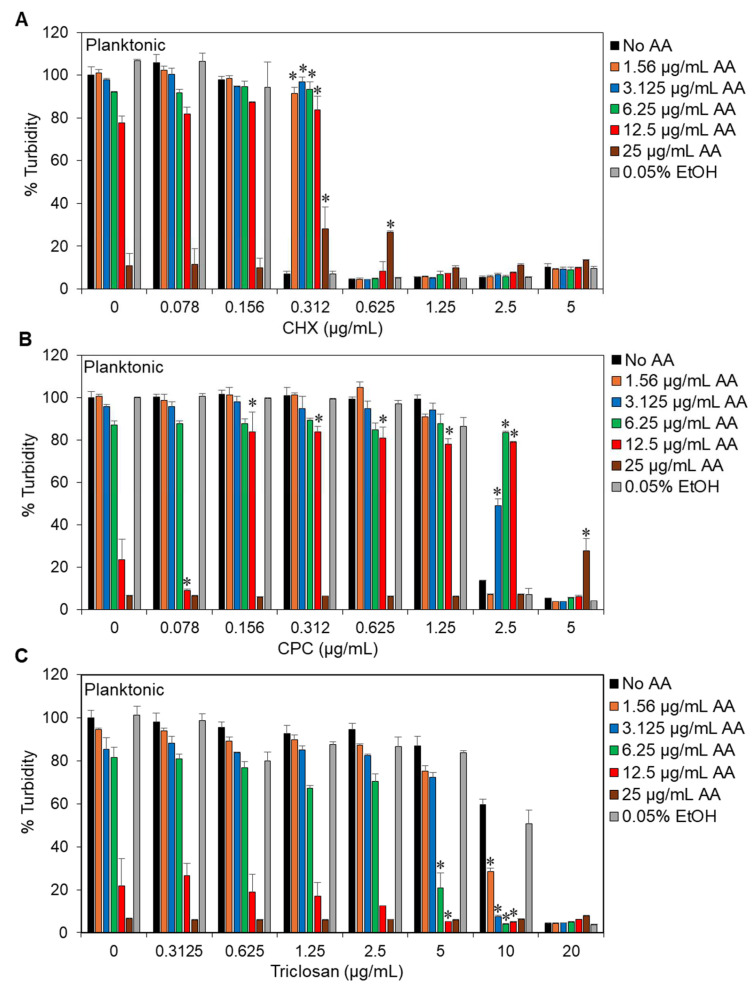
The effect of arachidonic acid on the anti-bacterial activity of commonly used antiseptics against *S. mutans*. (**A**–**C**). Checkerboard assay of arachidonic acid (AA) with chlorhexidine (CHX; (**A**)), cetylpyridinium chloride (CPC; (**B**)), or triclosan (**C**). n = 3. * *p* < 0.05 when compared to single treatments. In (**A**,**B**), there was a significant antagonistic effect, while in C, there was a significant additive effect.

**Figure 3 antibiotics-13-00540-f003:**
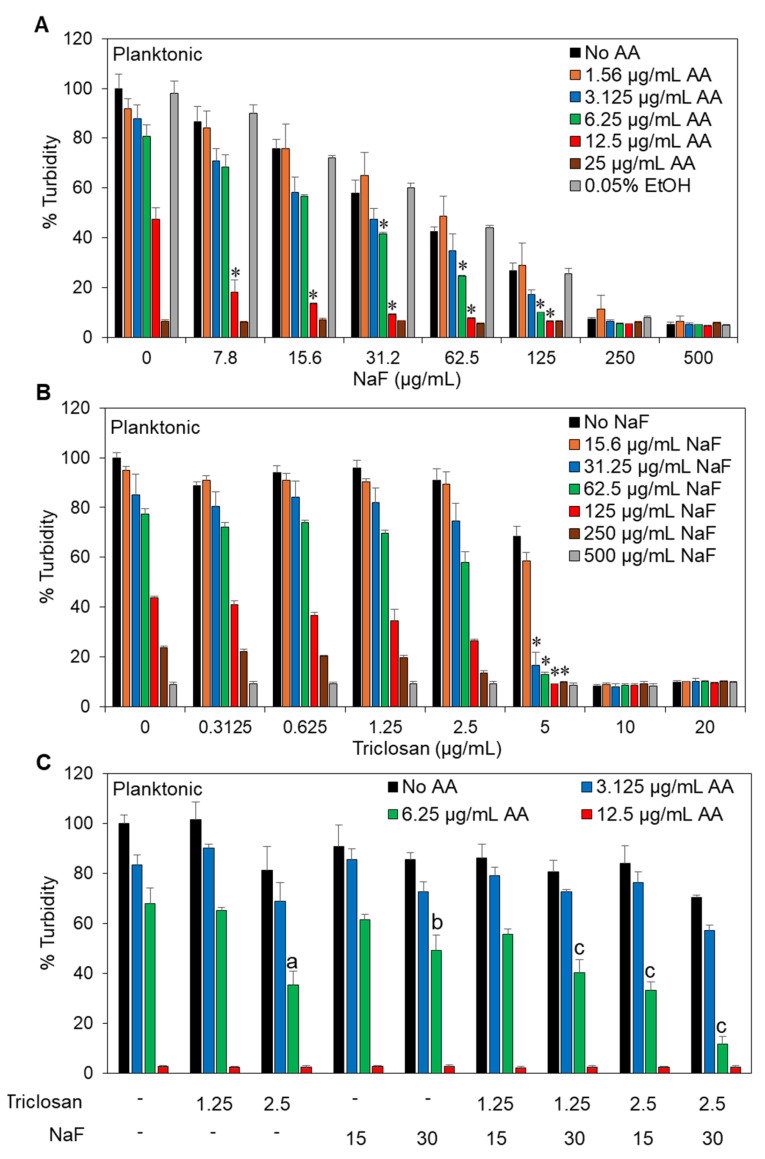
The addition of fluoride to the arachidonic acid and/or triclosan treatment increased the anti-bacterial effect against *S. mutans*. (**A**,**B**). The effect of sodium fluoride on the anti-bacterial activity of arachidonic acid (AA; (**A**)) or triclosan (**B**). n = 3. * *p* < 0.05 when compared to single treatments. (**C**). The anti-bacterial effect of various combinations of arachidonic acid (AA), triclosan, and sodium fluoride (NaF). n = 3. The letter “a” indicates *p* < 0.05 in comparison with individual AA and triclosan treatments. The letter “b” indicates *p* < 0.05 in comparison with individual AA and NaF treatments. The letter “c” indicates *p* < 0.05 in comparison with the combined triclosan/NaF treatment and 6.25 μg/mL AA alone.

**Figure 4 antibiotics-13-00540-f004:**
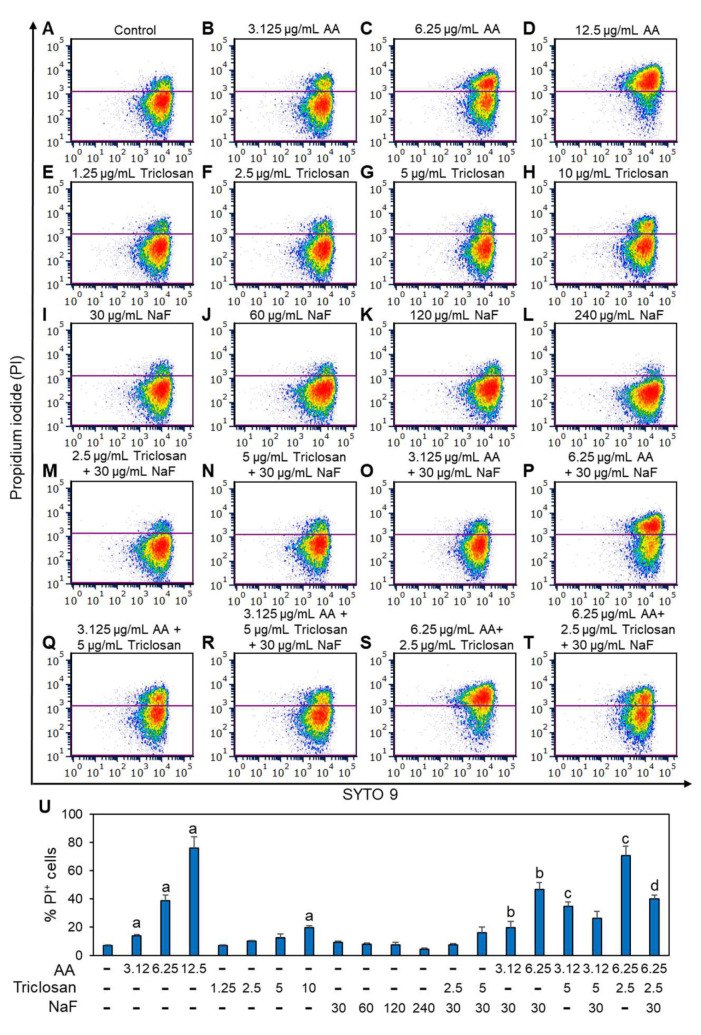
Live/dead SYTO 9/propidium iodide (PI) staining of planktonic-growing *S. mutans* that were exposed to various combinations of arachidonic acid (AA), triclosan, and/or sodium fluoride (NaF) for 2 h (**A**–**T**). PI vs. SYTO 9 dot plots of the different samples as indicated. (**U**). The percentage of PI-positive bacteria. n = 3. The letter “a” indicates *p* < 0.05 in comparison with control bacteria. The letter “b” indicates *p* < 0.05 in comparison with individual NaF and AA treatments. The letter “c” indicates *p* < 0.05 in comparison with individual triclosan and AA treatments. The letter “d” indicates *p* < 0.05 in comparison with the combined triclosan/NaF treatment and AA treatment alone.

**Figure 5 antibiotics-13-00540-f005:**
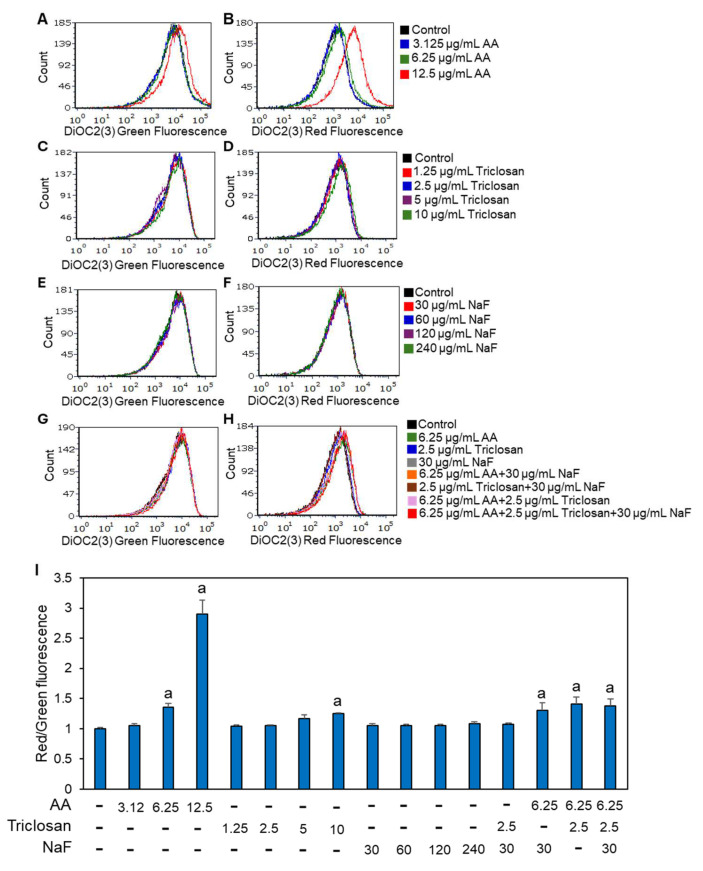
Potentiometric DiOC2(3) staining of planktonic-growing *S. mutans* that were exposed to various combinations of arachidonic acid (AA), triclosan, and/or sodium fluoride (NaF) for 10 min (**A**–**H**). Histograms of green fluorescence emitted by DiOC2(3) (**A**,**C**,**E**,**G**) vs. red fluorescence emitted by DiOC2(3) (**B**,**D**,**F**,**H**). (**I**). The ratio of red to green fluorescence for the different samples. n = 3. The letter “a” indicates *p* < 0.05 in comparison with control bacteria.

**Figure 6 antibiotics-13-00540-f006:**
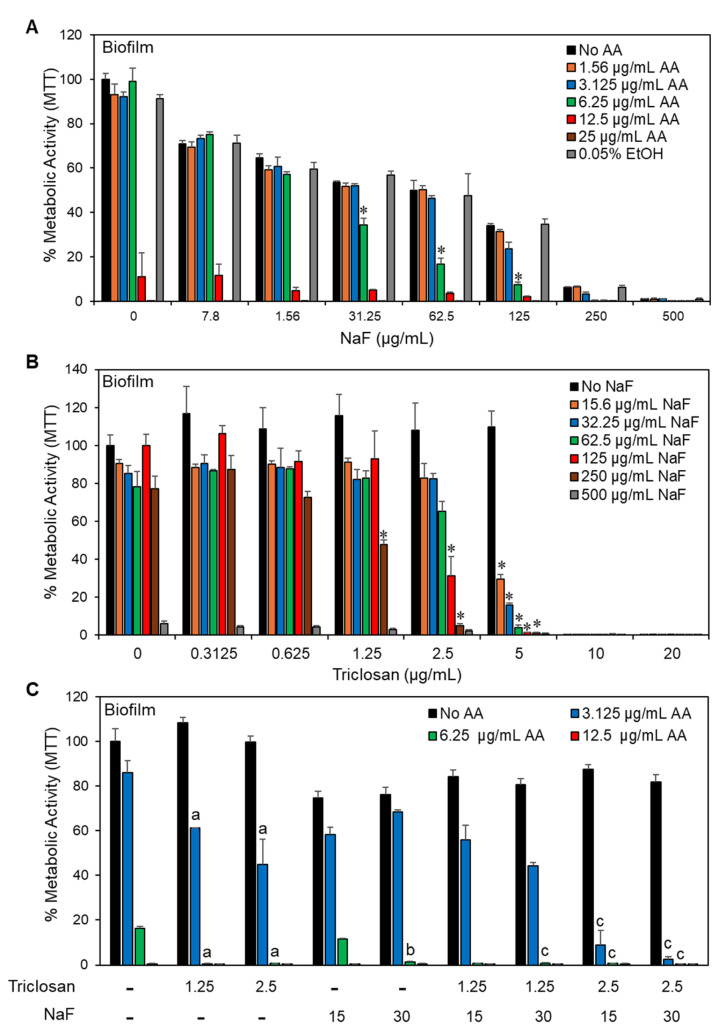
The effect of double and triple treatment on the metabolic effect of *S. mutans* biofilms. (**A**,**B**). The effect of sodium fluoride on the anti-biofilm activity of arachidonic acid (AA; (**A**)) or triclosan (**B**). n = 3. * *p* < 0.05 when compared with a single treatment. (**C**). The anti-biofilm effect of various combinations of arachidonic acid (AA), triclosan, and sodium fluoride (NaF). n = 3. The letter “a” indicates *p* < 0.05 in comparison with individual triclosan and AA treatments. The letter “b” indicates *p* < 0.05 in comparison with individual NaF and AA treatments. The letter “c” indicates *p* < 0.05 in comparison with the combined triclosan/NaF treatment and AA treatment alone.

**Figure 7 antibiotics-13-00540-f007:**
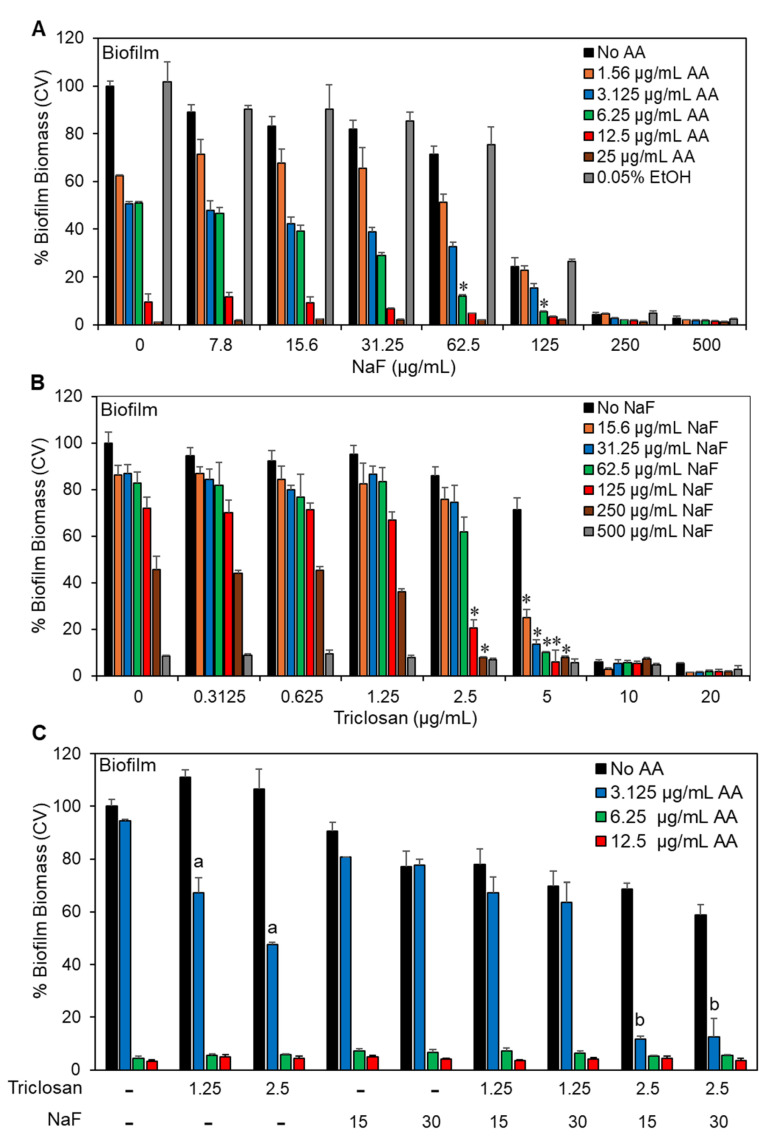
The effect of double and triple treatment on *S. mutans* biofilm biomass. (**A**,**B**). The effect of sodium fluoride on the anti-biofilm activity of arachidonic acid (AA; (**A**)) or triclosan (**B**). n = 3. * *p* < 0.05 when compared with single treatments. (**C**). The anti-biofilm effect of various combinations of arachidonic acid (AA), triclosan, and sodium fluoride (NaF). n = 3. The letter “a” indicates *p* < 0.05 in comparison with individual triclosan and AA treatments. The letter “b” indicates *p* < 0.05 in comparison with the combined triclosan/NaF treatment and AA treatment alone.

**Figure 8 antibiotics-13-00540-f008:**
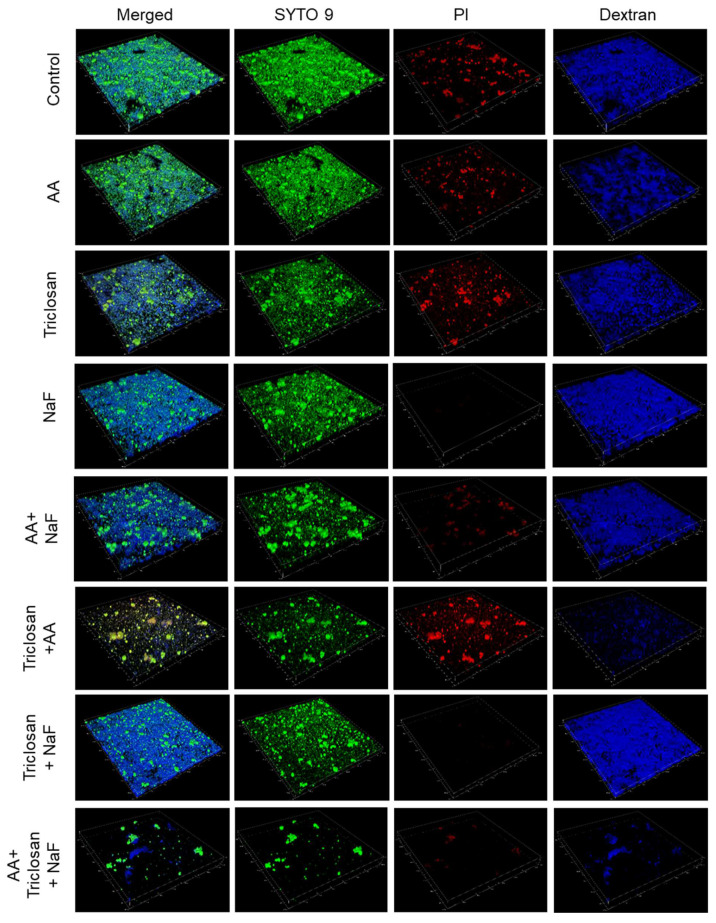
Three-dimensional images of confocal spinning disk microscopy of *S. mutans* biofilms exposed to various combinations of 3.125 μg/mL arachidonic acid (AA), 2.5 μg/mL triclosan, and 30 μg/mL NaF for 24 h and stained with SYTO 9 (green color), propidium iodide (PI, red color), and AlexaFluor^647^-conjugated Dextran 10,000 (blue color). The images cover a biofilm area of 1497.6 μm × 1497.6 μm.

**Figure 9 antibiotics-13-00540-f009:**
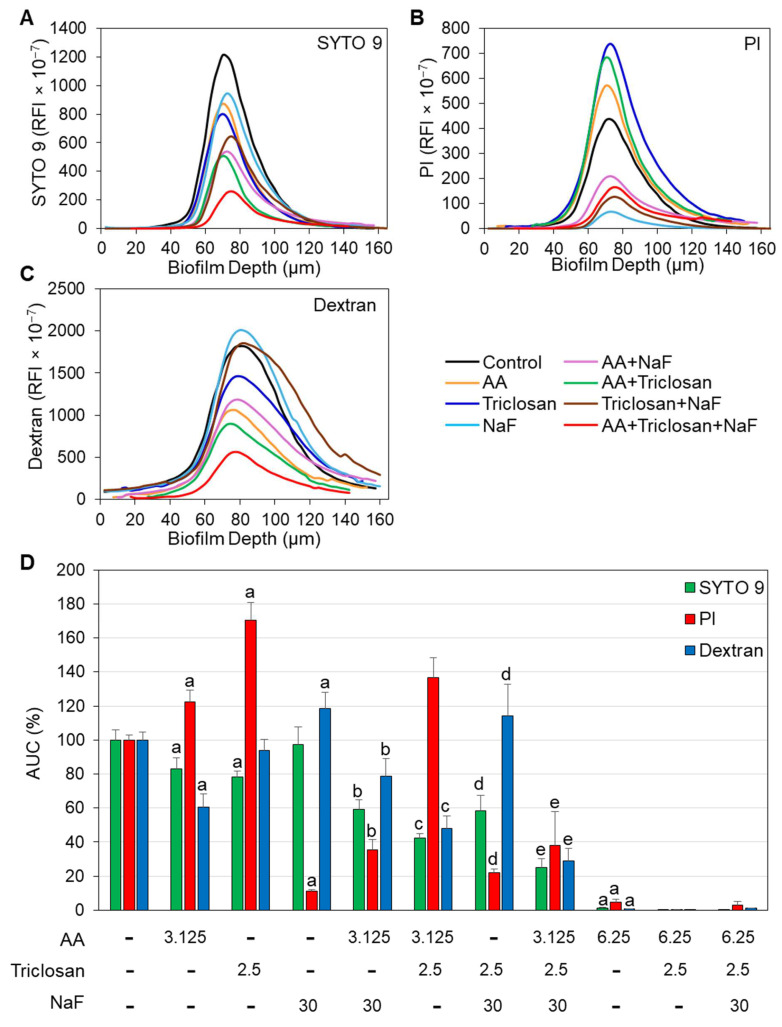
SYTO 9, PI, and Dextran 10,000 staining of the different biofilm layers by confocal spinning disk microscopy. (**A**–**C**). The relative fluorescence intensities (RFIs) of SYTO 9 (**A**), PI (**B**), and Dextran 10,000 (**C**) in the different biofilm layers. The curves are an average of 7–10 samples taken from different sites of 3–4 different samples. The numbers on the *Y*-axis are the calculated RFI values × 10^−7^. (**D**). The percentage area under the curve (AUC) of samples is presented in (**A**–**C**). The letter “a” indicates *p* < 0.05 in comparison with control bacteria. The letter “b” indicates *p* < 0.05 in comparison with individual NaF and AA treatment. The letter “c” indicates *p* < 0.05 in comparison with individual triclosan and AA treatments. The letter “d” indicates *p* < 0.05 in comparison with individual triclosan and NaF treatments. The letter “e” indicates *p* < 0.05 in comparison with the combined triclosan/NaF treatment and AA treatment alone.

**Figure 10 antibiotics-13-00540-f010:**
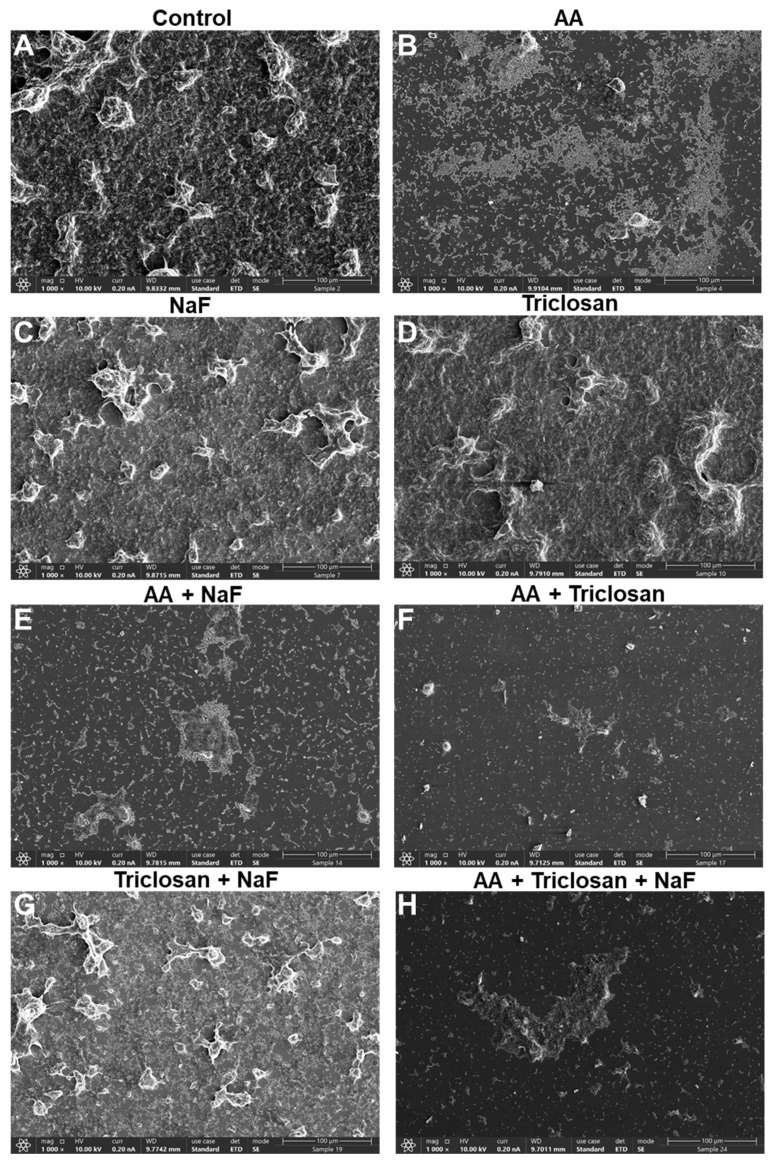
Panoramic HR-SEM images of *S. mutans* biofilms exposed to various combinations of 3.125 μg/mL arachidonic acid (AA), 2.5 μg/mL triclosan, and 30 μg/mL NaF for 24 h. (**A**) Control biofilm; (**B**) AA-treated biofilm; (**C**) NaF-treated biofilm; (**D**) Triclosan-treated biofilm. (**E**) AA/NaF-treated biofilm. (**F**) AA/triclosan-treated biofilm. (**G**) Triclosan/NaF-treated biofilm. (**H**) Triple-treated biofilm. Magnification × 1000. The bar represents 100 μm.

**Figure 11 antibiotics-13-00540-f011:**
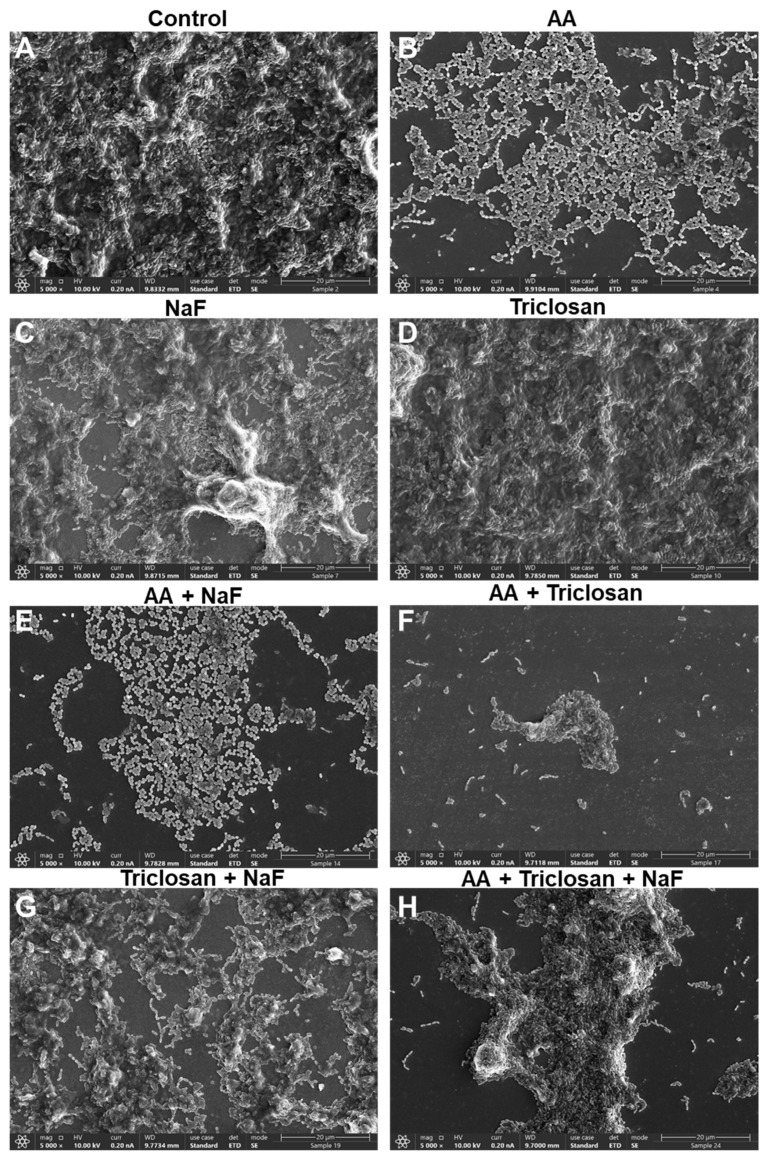
HR-SEM images of *S. mutans* biofilms exposed to various combinations of 3.125 μg/mL arachidonic acid (AA), 2.5 μg/mL triclosan, and 30 μg/mL NaF for 24 h. (**A**) Control biofilm; (**B**) AA-treated biofilm; (**C**) NaF-treated biofilm; (**D**) Triclosan-treated biofilm. (**E**) AA/NaF-treated biofilm. (**F**) AA/triclosan-treated biofilm. (**G**) Triclosan/NaF-treated biofilm. (**H**) Triple-treated biofilm. Magnification × 5000. The bar represents 20 μm. The images focus on biofilm-rich regions.

**Figure 12 antibiotics-13-00540-f012:**
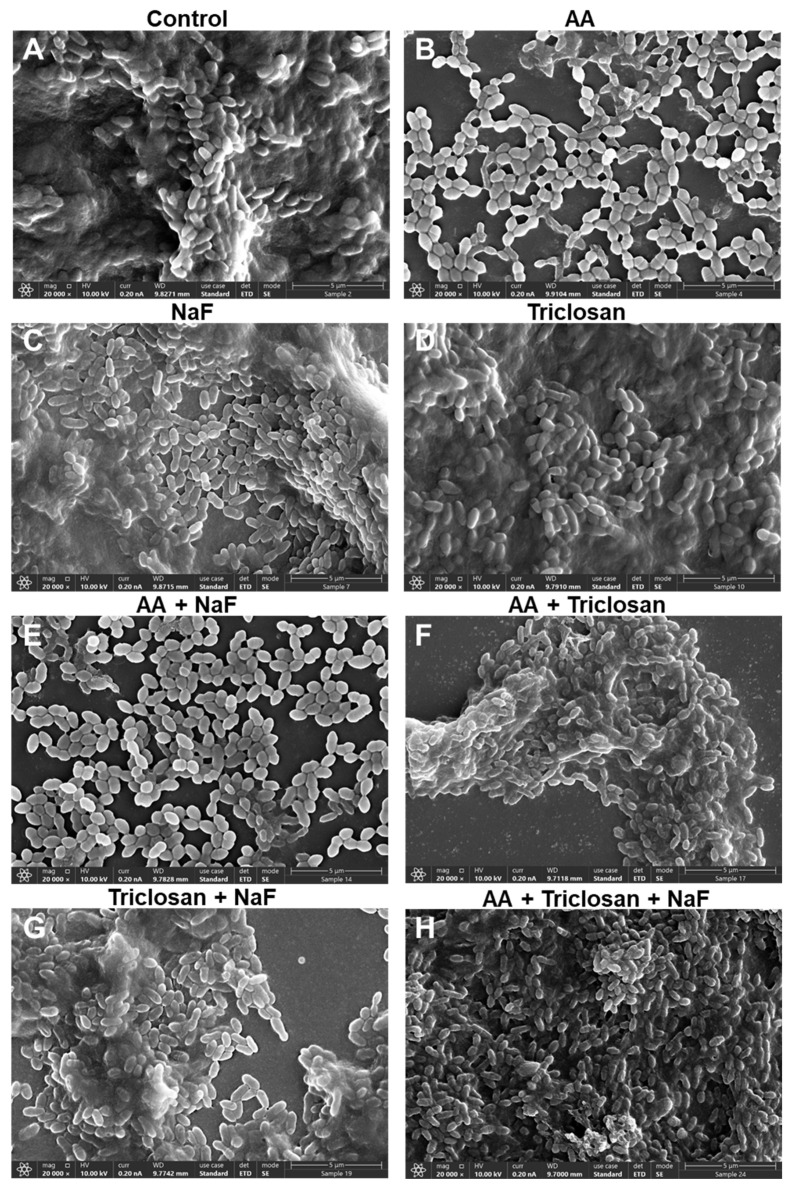
HR-SEM images of *S. mutans* biofilms exposed to various combinations of 3.125 μg/mL arachidonic acid (AA), 2.5 μg/mL triclosan, and 30 μg/mL NaF for 24 h. (**A**) Control biofilm; (**B**) AA-treated biofilm; (**C**) NaF-treated biofilm; (**D**) Triclosan-treated biofilm. (**E**) AA/NaF-treated biofilm. (**F**) AA/triclosan-treated biofilm. (**G**) Triclosan/NaF-treated biofilm. (**H**) Triple-treated biofilm. Magnification × 20,000. The bar represents 5 μm. The images focus on biofilm-rich regions.

## Data Availability

Raw data are available upon reasonable request.

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
