# Peer review of "Enhanced Anti-Bacterial Activity of Arachidonic Acid against the Cariogenic Bacterium Streptococcus mutans in Combination with Triclosan and Fluoride"

_antibiotics, 2024, doi:10.3390/antibiotics13060540_

Round 1

Reviewer 1 Report

Comments and Suggestions for Authors

A well-designed and very well-written manuscript. Thank you.

Only one minor comment:  For the viability assay, did the authors detect any eventual growth beyond the 24-hour test period, e.g. at 48, 72, 96 hours (or a week)?  Bacteria exposed to antimicrobial agents will go through a period of "non-viability" (usually within the test period) but in time, the survivors will adapt and grow.

Minor corrections:

1. Line 17 - "... anti-biofilm activities against this bacterium."

2. Line 39 - "... dental caries by virtue of its ability..."

3. Line 122 - Streptococcus pneumoniae (first use)

4. Line 123 - St. aureus (different genus abbreviation to differentiate between Streptococcus and Staphylococcus)

5. Line 450 - caries-inhibiting (hyphenated word)

Author Response

We would like to thank the Reviewer for critically reading our manuscript and providing constructive comments.

A well-designed and very well-written manuscript. Thank you.

Comment: Only one minor comment:  For the viability assay, did the authors detect any eventual growth beyond the 24-hour test period, e.g. at 48, 72, 96 hours (or a week)?  Bacteria exposed to antimicrobial agents will go through a period of "non-viability" (usually within the test period) but in time, the survivors will adapt and grow.

Response: We have studied planktonic growth for up to 48 h. There was no recovery of bacterial growth at this time point and similar results were obtained at 48 h compared to 24 h. In order to emphasize this issue, we have added the following text to the manuscript: "No bacterial regrowth was observed with these combinations even after a 48 h incubation (data not shown)."

Minor corrections:

  1. Line 17 - "... anti-biofilm activities against this bacterium."

Corrected.

  1. Line 39 - "... dental caries by virtue of its ability..."

Corrected.

  1. Line 122 - Streptococcus pneumoniae(first use)

Corrected.

  1. Line 123 - St. aureus (different genus abbreviation to differentiate between Streptococcus and Staphylococcus)

Corrected.

  1. Line 450 - caries-inhibiting (hyphenated word)

Corrected.

Reviewer 2 Report

Comments and Suggestions for Authors

This is a comprehensive study and a well written paper. The authors have been careful not to over-interpret their findings. Only some very minor revisions are needed.

Line 70 does not properly explain the effect of Zn or arginine. These are not rheological modifiers themselves. A better wording would be:

"Zinc ions and L-arginine have also been added to toothpaste. These reduced the amount of extracellular polymeric substances within the dental plaque biofilm, making it easier to remove by shear forces."

Line 103. There should be brief mention of allergies to CHX and resistance when discussion limitations of CHX. Both are well documented health issues.

Line 569. Mention that the cultures were gorown under static conditions (no agitation). This aspect is important as it influences S. mutans biofilm characteristics. 

The authors should also mention here or later that the culture plates used (ibiTreat) had a slightly negatively charged hydrophilic surface that facilitated bacterial cell attachment. Because of different substrates, biofilms formed on these would be slightly different to those formed on glass, which were used for creating SEM images.

Line 587. The correct word is ibiTreat (with a capital T), not ibitreat.

Author Response

Reviewer 2:

We would like to thank the Reviewer for critically reading our manuscript and providing constructive comments.

This is a comprehensive study and a well written paper. The authors have been careful not to over-interpret their findings. Only some very minor revisions are needed.

Comment: Line 70 does not properly explain the effect of Zn or arginine. These are not rheological modifiers themselves. A better wording would be:

"Zinc ions and L-arginine have also been added to toothpaste. These reduced the amount of extracellular polymeric substances within the dental plaque biofilm, making it easier to remove by shear forces."

Response: We have accordingly rephrased the text: "Zinc ions and L-arginine have also been added to toothpaste. These compounds reduce the amount of extracellular polymeric substances within the biofilm, making it easier to remove the dental plaque by shear forces ".

Comment: Line 103. There should be brief mention of allergies to CHX and resistance when discussion limitations of CHX. Both are well documented health issues.

Response: We have accordingly added these limitations of chlorhexidine. The following text has been added: "Other limitations of CHX may be acquisition of resistance mechanisms [27,33], allergic reactions and severe anaphylactic shock in rare cases [34]."

Comment: Line 569. Mention that the cultures were grown under static conditions (no agitation). This aspect is important as it influences S. mutans biofilm characteristics. 

Response: We have accordingly rephrased the sentence to: "After a 24 h incubation under static conditions".

Comment: The authors should also mention here or later that the culture plates used (ibiTreat) had a slightly negatively charged hydrophilic surface that facilitated bacterial cell attachment. Because of different substrates, biofilms formed on these would be slightly different to those formed on glass, which were used for creating SEM images.

Response: We have accordingly added the following text to Discussion: "Three different surfaces were used for biofilm formation: Tissue culture grade polystyrene wells with a negatively charged hydrophilic surface, ibiTreat slides with a slightly negatively charged hydrophilic surface, and uncharged glass pieces. Although the structure of the biofilm is influenced by the surface properties, the increased anti-biofilm effect of the triple treatment was observed regardless of the type of surface."

Comment: Line 587. The correct word is ibiTreat (with a capital T), not ibitreat.

Corrected.